# Causal Associations of Circulating Lipids with Osteoarthritis: A Bidirectional Mendelian Randomization Study

**DOI:** 10.3390/nu14071327

**Published:** 2022-03-22

**Authors:** Hongen Meng, Li Jiang, Zijun Song, Fudi Wang

**Affiliations:** The Fourth Affiliated Hospital, School of Public Health, Institute of Translational Medicine, Zhejiang University School of Medicine, Hangzhou 310058, China; hongenm@zju.edu.cn (H.M.); lijiang87@zju.edu.cn (L.J.); xtlyszj@zju.edu.cn (Z.S.)

**Keywords:** circulating lipid, apolipoprotein B, low-density lipoprotein cholesterol, mendelian randomization, osteoarthritis

## Abstract

Osteoarthritis (OA) imposes an increasing social burden due to global activity limitations, especially among the aged. Links between circulating lipids and OA have been reported; however, confounding data from observational studies have hindered causal conclusions. We used Mendelian randomization (MR) approach to evaluate the genetic causal effects of circulating apolipoproteins and lipoprotein lipids on OA risk. Genetic instruments at the genome-wide significance level (*p* < 5 × 10^−8^) were selected from genome-wide association studies (*n* = 393,193–441,016 individuals). Summary-level OA data were obtained from the UK Biobank (39,427 cases, 378,169 controls). Bidirectional two-sample Mendelian randomization (MR) analyses used MR-Egger, weighted median, and MR-PRESSO for sensitivity analysis. Genetic predisposition to 1-SD increments of Apolipoprotein B (APOB), and low-density lipoprotein (LDL) was associated with a decreased risk of knee or hip OA (KHOA) (odds ratio (OR) = 0.925, 95% confidence interval (95% CI): 0.881–0.972, *p* = 0.002; OR = 0.898, 95% CI: 0.843–0.957, *p* = 0.001) and hip OA (HOA) (OR = 0.894; 95% CI: 0.832–0.961, *p* = 0.002; OR = 0.870 95% CI: 0.797–0.949, *p* = 0.002). Genetically predicted APOB showed an association with knee OA (KOA) (OR per SD increase, 0.930, 95% CI: 0.876–0.987, *p* = 0.016). The OR of KOA was 0.899 (95% CI: 0.835–0.968, *p* = 0.005) for a 1-SD increase in LDL. Apolipoprotein A1, high-density lipoprotein, and triglycerides showed no association. Inverse MR showed no causal effect of KOA, HOA, or KHOA on these serum lipids. Distinct protective genetic-influence patterns were observed for APOB and LDL on OA, offering new insights into relationships between lipids and OA risk and a better understanding of OA etiology.

## 1. Introduction

Osteoarthritis (OA) is the main cause of activity limitation in the elderly, especially in older women [1]. It is a chronic degenerative joint disease (occurring preferentially in the hands, knees, hips, and feet) or a wear-and-tear disorder. Globally, more than 500 million people suffer from this disease, accounting for 7% of the world population [2], and this is expected to increase to 15–20% of the population by 2050, according to the United Nations [3]. Until now, no effective treatments have been available for OA apart from end-stage surgeries, such as total joint arthroplasty. However, the medical costs of surgeries are increasing [4], and OA takes a considerable personal, economic, and societal toll. Therefore, clarifying the causes and early-stage management of this tricky disease is very important.

OA is a complicated disease that is caused by many genetic, environmental, and systemic factors, including aging, gender, race, inflammation, and metabolism [5,6]. The concept of metabolic OA has recently been put forward, with a crucial role for metabolic disturbances and interactions proposed for the onset and development of OA [6,7]. Gkretsi et al. were the first to report a correlation between abnormal high-density cholesterol (HDL) metabolism and OA pathogenesis through epidemiological studies that revealed a link between increased cholesterol levels and the risk of knee, hip, and/or generalized OA [8]. An association between serum cholesterol levels and OA was subsequently reported with hand OA in women [9] and with knee OA [10]. Nevertheless, many studies have presented contradictory results [11,12], including the lack of a significant correlation between serum cholesterol and radiographic knee OA in an early study from Gothenburg [11], and no association between plasma low density cholesterol (LDL) or low HDL and the radiographic or symptomatic OA risks in the Multicenter Osteoarthritis Study (MOST) cohort [12]. By contrast, two recent large meta-analyses studies have shown a strong relationship between dyslipidemia and OA [13,14]. This unclear relationship between lipids and OA necessitates further clarification of the causal effects of serum lipids on OA.

Mendelian randomization (MR) is a new strategy for investigating causation between different traits based on Mendel’s laws of inheritance. MR uses genetic predisposition to avoid biases, such as sample size and short follow-up duration, in observational studies. At present, only one study has explored the effects of obesity-related risk factors, including body mass index (BMI), bone mineral density, serum HDL and LDL, triglyceride (TG) levels, type 2 diabetes, systolic blood pressure (BP), and C-reactive protein levels, on OA incidence, and the authors claimed no evidence of causality for HDL, LDL, and TG on knee OA (KOA), hip OA (HOA), or hand OA [15]. In the same year, another related study examined the effects of cardiometabolic risk factors (LDL, HDL, TG, BMI, fasting plasma glucose, and BP) on OA and reported protective effects of LDL on OA but no strong evidence for any effects of HDL and TG on OA [16]. Another study also reported protective effects of high LDL on OA risk [17]. However, the genetic instrument variables (IVs) used were all obtained from selected genome-wide association studies (GWASs) conducted in 2013 or even before.

In the present study, we conducted a bidirectional two-sample MR study and focused on serum lipoproteins and lipids, including Apolipoprotein A1 (APOA1), Apolipoprotein B (APOB), HDL, LDL, and TG. Our study was based on the latest databases and explored the correlation between these lipid factors and the risk of OA (KOA, HOA, and OA of the knee or hip (KHOA)).

## 2. Materials and Methods

### 2.1. Study Design and Data Sources

The study design overview and the assumptions of the MR study are shown in Appendix A. The genetic instruments for the exposures were obtained from published GWASs [18]. Data for OA were obtained from the UK Biobank [19]. Detailed information about the data sources used is displayed in Table 1. All studies had been approved by a relevant ethical review board, and all participants had given informed consent.

### 2.2. Genetic Instrument Selection

Single nucleotide polymorphisms (SNPs) associated with APOA1, APOB, HDL, LDL, and TG at the genome-wide level were obtained from large GWAS’ [18]. The mean age of participants was 56.9 y (range 39–73 y) and 54.2% of them were women. Detailed information about the GWASs used is presented in Table 1. SNPs without linkage disequilibrium (defined by r^2^ < 0.001 and clump distance >1 Mb) were used as IVs. All selected SNPs for APOA1, APOB, HDL, LDL, and TG are listed in Appendix A.

### 2.3. Data Sources for OA

Summary-level data for the associations of exposure-associated SNPs with OA were available from the UK Biobank study [19], with up to 39,427 cases and 378,169 controls of European ancestry. The OA cases were from Arthritis Research UK Osteoarthritis Genetics (arcOGEN, which is a collection of unrelated UK-based individuals of European ancestry with knee and/or hip osteoarthritis from the arcOGEN Consortium) and were ascertained based either on clinical evidence of disease to a level requiring joint replacement or on radiographic evidence of disease (Kellgren–Lawrence grade ≥ 2). The controls were from the United Kingdom Household Longitudinal Study (UKHLS), which is a longitudinal panel survey of 40,000 UK households (England, Scotland, Wales, and Northern Ireland) representative of the UK population. Detailed information regarding the SNPs of the 5 lipids used as IVs to conduct this MR analysis is displayed in Appendix A.

All studies included in the GWASs had been approved by relevant ethical review committees, and participants provided written informed consent. The current study only used summary-level data that was publicly available. Thus, no additional ethical review was required for this study.

### 2.4. Statistical Analysis

We used the multiplicative random-effects inverse variance weighted (IVW) model as the main statistical method. Three sensitivity analyses, namely the weighted median [20], MR-Egger [21], and MR-PRESSO [22], were performed to examine the consistency of associations and to detect possible pleiotropy. The weighted median method provides consistent causal estimates when more than 50% of the weight comes from valid instruments [20]. The MR-Egger regression can detect pleiotropy by its intercept and provide estimates after the correction for pleiotropy; however, it compromises statistical power [21]. The MR-PRESSO method can detect outliers and provide causal estimates after the removal of identified outliers [22]. The distortion test embedded in the MR-PRESSO analysis can distinguish the difference between the estimates before and after outlier removal [22]. Cochrane’s Q value was used to assess the heterogeneity among SNP estimates in each analysis. The horizontal heterogeneity effect was examined by the IVW test and the MR-Egger regression. A leave-one-out sensitivity analysis was also performed to monitor whether significant associations were dominated by a single SNP. Associations with a *p*-value < 0.01 (0.05/5 exposures) were deemed significant associations, and associations with a *p*-value > 0.01 and <0.05 were regarded as suggestive associations. Statistical powers were calculated using the mRnd power calculation tool, assuming the true causal OR was 1.10 [23]. We performed the Mendelian randomization analysis using the Two-Sample MR package, an R package, and a GWAS summary data library developed as a platform for performing Mendelian randomization tests and sensitivity analyses [24,25]. All the statistical analyses were performed using R (version 4.1.1) software (TwoSampleMR [24], MR-PRESSO [22], and MendelianRandomization [26] packages).

## 3. Results

Genetic liability due to APOB showed an association with a decreased risk of KHOA and HOA and a suggestive association with a decreased risk of KOA (*p* = 0.002, *p* = 0.002, and *p* = 0.016, respectively) (Table 2). These associations remained significant in the other sensitivity analyses. For a 1 SD increment of APOB, the odds ratio (OR) and 95% confidence intervals (CI) for KHOA, KOA, and HOA were OR = 0.925 (95% CI: 0.881–0.972), OR = 0.930 (95% CI: 0.876–0.987), and OR = 0.894 (95% CI: 0.832–0.961), respectively.

The genetic predisposition to higher LDL was associated with a decreased risk of KHOA (OR = 0.898, 95% CI: 0.843–0.957, *p* = 0.001) (Table 3). The associations remained consistent in the sensitivity analyses using the weighted median and MR-Egger methods (Table 3). Five outliers were detected in the MR-PRESSO analyses; however, the associations persisted after the removal of these outliers (Table 3). Only limited data supported associations of genetically predicted LDL with KOA (OR = 0.899, 95% CI: 0.835–0.968, *p* = 0.005) or HOA (OR = 0.870, 95% CI: 0.797–0.949, *p* = 0.002) (Table 3). For KOA, the protective effect persisted in the sensitivity analyses in the MR-Egger methods and MR-PRESSO analyses after the removal of five outliers, but not in the weighted median methods. For HOA, the negative association was borderline in the weighted median and MR-Egger methods but became significant in MR-PRESSO analyses after the removal of five outliers. No associations were detected for APOA1, HDL, and TG with OA in the primary analysis or in the sensitivity analyses of each data source (Appendix A).

The intercept in the MR-Egger regression indicated no pleiotropy in any analysis (all *p* values > 0.05) (Table 2 and Table 3, Appendix A). No clear violations of the MR assumptions were indicated after generating forest, funnel, and scatter plots (Appendix A).

An inverse MR of OA on serum lipids was similarly conducted, except the Bonferroni-adjusted *p* < 0.05/3 = 0.017 was used. No causal effect was detected for KOA, HOA, or KHOA on serum lipids (APOA1, APOB, HDL, LDL, and TG) using 7 IVs of KOA, 23 IVs of HOA, 22 IVs of KHOA as exposures and serum lipids as outcomes, through multiple MR sensitivity analysis (Appendix A).

## 4. Discussion

The multiple MR sensitive analyses led to the conclusion that circulating APOB was negatively associated with the risk of KHOA and HOA, and that serum LDL was negatively associated with the risk of KHOA, KOA, and HOA. No other genetic relationships were detected between other lipids and the risk of KOA, HOA, or KHOA, nor was there a genetic relationship between KOA, HOA, or KHOA and these five lipids.

This study is the first one to demonstrate a protective effect of APOB on KOA, HOA, and KHOA, despite previous demonstrations of no or positive correlations with OA risks in a number of studies. For example, Zhang et al. found no significant differences in the APOB levels in the serum and synovial fluid of primary KOA patients or healthy individuals [27]. In a mouse model of feeding a cholesterol-rich diet, APOB accumulation in synovial macrophages did not affect the thickening of the synovial lining or induce cartilage damage [28]. Similarly, the mutations in APOB (rs693 and rs1042031) increased the risk of steroid-induced osteonecrosis of the femoral head [29]. However, an osteochondrosis study of horses determined that APOB and APOB-3G (similar to Apolipoprotein B mRNA editing enzyme catalytic polypeptide-like 3G) from leukocytes were relatively under-expressed in the osteochondrosis-affected group, indicating an inverse correlation between APOB and osteochondrosis which culminated in OA [30]. As the apolipoprotein of LDL and the ligand for the LDL receptor, the protective effects of APOB on OA might be mediated through LDL, which was negatively associated with the risk of KHOA, KOA, and HOA in this study.

LDL has also shown an association with OA. One comparative study showed that the mean level of LDL was significantly (*p* < 0.01) higher in KOA [31], in agreement with another study of primary KOA [22]. A case-control study showed that LDL was independently associated with the radiographic severity of KHOA [7]. An experimental study showed that high LDL participated in the progression of KOA through synovial inflammation and ectopic bone formation [32]. Many other studies have supported an association between the oxidation of low-density lipoprotein (ox-LDL) and OA [33,34,35]. However, de Seny et al. found that LDL could decrease APOA1 levels and serum amyloid A protein-induced joint inflammation in human primary chondrocytes and fibroblast-like synoviocytes [36]. A recent MR study using the Malmö Diet and Cancer Study cohort [17] also claimed that LDL levels show a genetically inverse relationship to OA risks, which supports our present findings. Whether the simultaneous elevation of APOB and LDL exerts a protective effect is unclear and further studies are needed.

Some epidemiological and experimental studies have also concluded that no relationship exists between OA and either APOA1, HDL, or TG. One previous study reported no significant difference in serum APOA1 levels between OA patients and healthy controls [37]. Schwager et al. also found no significant association between HDL and OA based on the Multicenter Osteoarthritis Study (MOST) cohort [12]. Genetic causality between HDL and TG with OA was also absent in a UKBB based MR study [15]. The use of a statin (a serum-lipid lowering drug) had no impact on KOA progression [38], implying no effect of HDL and TG on OA.

The underlying mechanisms of the protective effect of APOB and LDL on OA remain to be explored through additional research. APOB in the synovial fluid appears to be important in the response to the TNFα-NFκB inflammatory signal through alpha-enolase (ENO1), independent of the LDL receptor [1]. Much research is now focused on the effect of ox-LDL on OA. The lectin-like oxidized low density lipoprotein receptor 1 (ox-LDL/LOX1) system might be involved in the progression of OA through vascular endothelial growth factor (VEGF)/ peroxisome proliferators-activated receptor γ (PPAR-γ) [35], monocyte chemoattractant protein 1 (MCP-1) [33], IL-1β [39], and ROS [40,41]. A recent study reported that feeding a serum LDL-increasing western diet to a mouse model of KOA did not play a role in the Nox isoform 2 (Nox2)-mediated ROS production or the IL-1β-mediated pro-inflammatory effect [42,43], indicating that LDL might not be deleterious for OA. Consistent with this, a high-fat diet intake in healthy and OA rabbits did not change the expression of pro-inflammatory and catabolic (IL-1β, IL-6, matrix metallopeptidase 13 (MMP-13), MCP1, or cyclooxygenase-2 (COX-2)) genes or proteins, nor did it modify the cartilage structure. Further, in cultured human OA articular chondrocytes, ox-LDL did not affect chondrocyte viability in the presence or absence of IL-1β and TNFα pro-inflammatory stimuli and even decreased the induction of IL-1β, IL-6, MCP-1, MMP-13, iNOS, and COX-2 gene expression increased MMP-13 and COX-2 protein levels [44]. Ascone et al. also showed that the protective effects of LDL on OA arose due to attenuation of the onset of inflammation and cartilage destruction through Fc gamma Receptors (FcγRs) [45]. Further studies are warranted to identify the potential mechanisms by which APOB and LDL protect against the occurrence and progression of OA.

To the best of our knowledge, this study is the largest genetic correlation study of the effects of serum APOA1, APOB, HDL, LDL, and TG levels on KOA, HOA, and KHOA using MR analyses based on a bidirectional two-sample MR approach. The datasets employed were the most recent and largest, and both came from UKBB; therefore, the cohorts share the same European descent, and this reduces the bias caused by population stratification. The use of strict SNP selection terms, with LD < 0.001 and *p* < 0.01, and choice of SNPs at locations ≥1 MB distant from each other provided further increases in precision and statistical power. In addition to using MR-Egger, weighted median, IVW, and MR-PRESSO (raw, corrected outliers) methods in the MR sensitivity analysis, we also employed an inverse MR sensitivity analysis of OA on lipids to minimize the bias and provide a strong causal result. A few studies have previously shown a correlation between APOB or LDL and OA, but we have further analyzed the genetically correct effect to rule out false-positive results.

Despite all these efforts, the present study still has some limitations. One is that only European descent participants were included, but African Americans have a high incidence of both KOA and HOA [46,47]. A second limitation is that only APOA1, APOB, HDL, LDL, and TG were examined. More genetic IVs of other lipid components should be taken into account to draw more accurate conclusions. Another limitation is that, even with the use of the weighted median method, bias caused by heterogeneity cannot be fully avoided. Moreover, although our reverse MR analysis found no significant causal evidence, the effect of OA on blood lipids requires further investigation based on independent GWAS and large prospective studies. Nonetheless, our study offers new insights into the relationships between lipids and the risk of OA and provides a better understanding of OA etiology.

## 5. Conclusions

A genetic protective effect was found for APOB and LDL on OA based on bidirectional MR analysis. Specifically, increased APOB was negatively related to the risks of KOA, HOA, and KHOA, and elevated LDL decreased the risks of KHOA.

## Figures and Tables

**Table 1 nutrients-14-01327-t001:** Characteristics of UK Biobank datasets.

Exposures		Consortium	No. SNPs	Sample Size		Adjustments	Population
	APOA1	UK Biobank	299	393,193		Age, sex, and genotyping chip array	European
	APOB	UK Biobank	198	439,214			
	HDL	UK Biobank	362	403,943			
	LDL	UK Biobank	177	440,546			
	TG	UK Biobank	313	441,016			
**Main Outcomes**		**Dataset**	**No. Cases**	**Control**	**Total**	**Adjustments**	**Population**
	KHOA	UK Biobank	39,427	378,169	417,596	Age, sex, genotyping chip array, and 10 genetic principal components	European
	KOA	UK Biobank	24,955	378,169	403,124		
	HOA	UK Biobank	15,704	378,169	393,873		

Abbreviations: APOA1, Apolipoprotein A1; APOB, Apolipoprotein B; HDL, high-density lipoprotein cholesterol; LDL, low-density lipoprotein cholesterol; TG, triglycerides; SNP, single-nucleotide polymorphism; KHOA, Osteoarthritis of knee or hip; KOA, Knee OA; HOA, Hip OA.

**Table 2 nutrients-14-01327-t002:** Associations of genetically predicted APOB with OA risks in MR analyses.

Main Outcome	Method	No. of SNPs	OR (95% CI)	P for Association	P for Heterogeneity Test	P for MR-Egger Intercept	P for MR-PRESSO Global Test	Statistical Power
KHOA	IVW	188	0.925 (0.881–0.972)	0.002	8.23 × 10^−12^	0.09		1.00
	MR Egger	188	0.889 (0.832–0.950)	0.001	2.21 × 10^−11^			
	Weighted median	188	0.900 (0.848–0.956)	0.001				
	MR-PRESSO (raw,3outliers)	185	0.927 (0.924–0.930)	0.001			<1 × 10^−4^	
KOA	IVW	188	0.930 (0.876–0.987)	0.016	3.81 × 10^−11^	0.19		1.00
	MR Egger	188	0.896 (0.827–0.972)	0.009	5.79 × 10^−11^			
	Weighted median	188	0.892 (0.824–0.966)	0.005				
	MR-PRESSO (raw,2outliers)	186	0.927 (0.923–0.930)	0.006			<1 × 10^−4^	
HOA	IVW	188	0.894 (0.832–0.961)	0.002	5.05 × 10^−9^	0.44		0.96
	MR Egger	188	0.871 (0.789–0.961)	0.006	4.79 × 10^−9^			
	Weighted median	188	0.873 (0.800–0.953)	0.002				
	MR-PRESSO (raw,4outliers)	184	0.881 (0.877–0.885)	<0.001			<1 × 10^−4^	

Abbreviations: APOB, Apolipoprotein B; OA, Osteoarthritis; KHOA, Osteoarthritis of the knee or hip; KOA, Knee OA; HOA, Hip OA; IVW, multiplicative random-effects inverse variance-weighted; SNP, single-nucleotide polymorphism; OR, odds ratio; CI, confidence interval.

**Table 3 nutrients-14-01327-t003:** Associations of genetically predicted LDL with OA risks in MR analyses.

Main Outcome	Method	No. of SNPs	OR (95% CI)	P for Association	P for Heterogeneity Test	P for MR-Egger Intercept	P for MR-PRESSO Global Test	Statistical Power
KHOA	IVW	162	0.898 (0.843–0.957)	0.001	3.24 × 10^−18^	0.18		1.00
	MR Egger	162	0.856 (0.778–0.941)	0.002	6.80 × 10^−18^			
	Weighted median	162	0.867 (0.810–0.927)	<0.001				
	MR-PRESSO (raw, 5outliers)	157	0.901 (0.897–0.905)	<0.001			<1 × 10^−4^	
KOA	IVW	163	0.899 (0.835–0.968)	0.005	2.99 × 10^−14^	0.25		0.97
	MR Egger	163	0.857 (0.767–0.957)	0.007	4.22 × 10^−14^			
	Weighted median	163	0.934 (0.854–1.021)	0.135				
	MR-PRESSO (raw, 5outliers)	158	0.900 (0.896–0.905)	0.002			<1 × 10^−4^	
HOA	IVW	163	0.870 (0.797–0.949)	0.002	1.07 × 10^−9^	0.56		0.88
	MR Egger	163	0.845 (0.741–0.963)	0.012	9.06 × 10^−10^			
	Weighted median	163	0.891 (0.798–0.994)	0.039				
	MR-PRESSO (raw, 5outliers)	158	0.863 (0.858–0.868)	<0.001			<1 × 10^−4^	

Abbreviations: LDL, low-density lipoprotein cholesterol; OA, Osteoarthritis; KHOA, Osteoarthritis of the knee or hip; KOA, Knee OA; HOA, Hip OA; IVW, multiplicative random-effects inverse variance-weighted; SNP, single-nucleotide polymorphism; OR, odds ratio; CI, confidence interval.

## Data Availability

All data are available in the submitted manuscript or related sources described in the manuscript. Appendix A were hosted on the Figshare website and cited as Appendix A. Data described in the manuscript, code book, and analytic code will be made available upon request pending.

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
