# Peer review of "Causal Associations of Circulating Lipids with Osteoarthritis: A Bidirectional Mendelian Randomization Study"

_nutrients, 2022, doi:10.3390/nu14071327_

Round 1

Reviewer 1 Report

The content of the manuscript matches well with the journal`s topic. The study supports the novel concept of "metabolic OA". The manuscript analysed the interrelation between OA (hip and/or knee) and occurrence of SNPs affecting serum lipid levels. The authors found that genetics for elevated APOB and LDL could impair the risk of OA. It is well structured.

introduction

were multiple nucleotide polymorphisms excluded from the study?

fist sentence could be supported by a reference

line 38: write simply "joint arthroplasty" since the manuscripts deals with knee and hip OA.

table 1: the adjustments could be better explained. How many female/male participiants? how about the age structure? What means "chip"?

discussion

"APOB and LDL " is there any hypotheses: how do elevated levels of both protect from OA onset?

line 192: "arthritis" which type?

line 194: osteonecrosis is a different entity

line 205: association with which type of OA? also line 208: "severity" of which OA? line 209: progression of which OA?

line 233: do not write "Mediated" with capital letter

line 236: introduce the abbreviations

limitations: "the effects of AO cannot be completely excluded" how does OA affect the occurence of genetic changes (SNPs)? how can it be explained?

Author Response

Response to Reviewer 1 Comments

Comments and Suggestions for Authors: The content of the manuscript matches well with the journal`s topic. The study supports the novel concept of "metabolic OA". The manuscript analysed the interrelation between OA (hip and/or knee) and occurrence of SNPs affecting serum lipid levels. The authors found that genetics for elevated APOB and LDL could impair the risk of OA. It is well structured.

Response: We appreciate the reviewer’s positive comments. Thank you very much for taking the time to review our work.

Point 1: were multiple nucleotide polymorphisms excluded from the study?

Response: Yes, as described in the Materials and Methods section (2.2. Genetic Instrument Selection) in Line 87, only Single nucleotide polymorphisms were derived as genetic instruments from selected genome-wide association studies (GWASs).

Point 2: fist sentence could be supported by a reference.

Response: We thank the reviewer for the valuable suggestions. The related reference has been added and incorporated into the revised manuscript (Line 33).

Related reference:

[1]Katz, J. N.; Arant, K. R.; Loeser, R. F. Diagnosis and Treatment of Hip and Knee Osteoarthritis: A Review. JAMA, 2021,325(6), 568–578.

Point 3: line 38: write simply "joint arthroplasty" since the manuscripts deals with knee and hip OA.

Response: We thank the reviewer’s constructive suggestion. The statement was corrected as suggested and has been incorporated into the revised manuscript (Line 38).

Point 4: table 1: the adjustments could be better explained. How many female/male participiants? how about the age structure? What means "chip"?

Response: The reviewer’s points are very well taken. For lipids-related GWAS, “The mean age of participiants was 56.9 y (range 39–73 y) and 54.2% of them were women”. This statement was added in the Materials and Methods section (2.2. Genetic Instrument Selection Line 88-89) and has been incorporated into the revised manuscript.

For the outcomes, the gender and age structure was not mentioned in the original article and related resources. We have already sent an email to the corresponding author Eleftheria Zeggini for a request for this information and haven't received a reply yet.

“chip” was described in the original article in the Supplementary section (Supplementary Figure 1: Flowchart depicting the two GWAS and their meta-analysis section). [Tachmazidou et al., Nature genetics, 2019]. “chip” might mean “genotyping chip array” in our understanding. This statement has been amended as “genotyping chip array” and has been incorporated into the revised manuscript (Table 1).

Related references:

[19]Tachmazidou, I.; Hatzikotoulas, K.; Southam, L.; Esparza-Gordillo, J.; Haberland, V.; Zheng, J.; et al.. Identification of new therapeutic targets for osteoarthritis through genome-wide analyses of UK Biobank data. Nat Genet 2019,51,230-236.

Point 5: "APOB and LDL " is there any hypotheses: how do elevated levels of both protect from OA onset?.

Response: This is a particularly good question. We have added this quetion in Line 212-213.

To the best of our knowledge, there are not any hypotheses about elevated levels of both APOB and LDL protecting from OA onset. Only de Seny et al. found that LDL could decrease APOA1 levels and serum amyloid A protein-induced joint inflammation in human primary chondrocytes and fibroblast-like synoviocytes (Line 208-210). Moreover, Ascone et al. showed that the protective effects of LDL on OA arose due to attenuation of the onset of inflammation and cartilage destruction through FcγRs (Line 240-242). They both have been discussed in the Discussion section (Line 208-210 and Line 240-242). 

Related references:

[37]de Seny, D.; Cobraiville, G.; Charlier, E.; Neuville, S.; Lutteri, L.; Le Goff, C.; et al.. Apolipoprotein-A1 as a damage-associated molecular patterns protein in osteoarthritis: ex vivo and in vitro pro-inflammatory properties. PLoS One 2015,10,e0122904.

[47]Ascone, G.; Di Ceglie, I.; van den Bosch, M.H.J.; Kruisbergen, N.N.L.; Walgreen, B.; Sloetjes, A.W.; et al.. High LDL-C levels attenuate onset of inflammation and cartilage destruction in antigen-induced arthritis. Clin Exp Rheumatol 2019,37,983-993

Point 6:

line 192: "arthritis" which type?

line 194: osteonecrosis is a different entity  

Response: We thank the reviewer for this constructive suggestion. In response, the type of  "arthritis" in line 192 is rheumatoid arthritis. As mentioned in line 194, both rheumatoid arthritis and osteonecrosis are different entities other than OA. We are sorry for the unclear clarification of arthritis and we have removed these two sentences.

Point 7:

line 205: association with which type of OA?

line 208: "severity" of which OA?

line 209: progression of which OA?

line 233: do not write "Mediated" with capital letter

line 236: introduce the abbreviations .

Response: We thank the reviewer for this constructive suggestion. In response, the clarification of which OA mentioned has been amended (Line 202, 204, 205, 231). The formatting errors have been corrected as suggested (Line 232). The abbreviations have been introduced (Line 228, 229, 231, 235, 242). All of these modifications have been incorporated into the revised manuscript.

Point 8: limitations: "the effects of OA cannot be completely excluded" how does OA affect the occurence of genetic changes (SNPs)? how can it be explained?.

Response: Thank you for raising this question. We are very sorry for the unclear statement, this sentence has been revised as "Moreover, although our reverse MR analysis found no significant causal evidence, the effect of OA on blood lipids requires further investigation based on independent GWAS and large prospective studies." (Line 264-266)

Reviewer 2 Report

The present study aimed to identify/validate risk factors for Osteoarthritits related to circulating lipoproteins. The study uses updated and large genetic platforms to perform a relevant and novel MR based analysis, with signficant findings for APOB that need further experimental validation. The main limitation of the study is the small number of variables analyzed and that there is not a validation cohort included. However, the study is of interest and generally well conducted. The significant number of typos and formating errors in the figures should be corrected for clarity sake.

Author Response

Comments and Suggestions for Authors

The present study aimed to identify/validate risk factors for Osteoarthritits related to circulating lipoproteins. The study uses updated and large genetic platforms to perform a relevant and novel MR based analysis, with signficant findings for APOB that need further experimental validation. The main limitation of the study is the small number of variables analyzed and that there is not a validation cohort included. However, the study is of interest and generally well conducted. The significant number of typos and formatting errors in the figures should be corrected for clarity sake.

Response: We thank the reviewer for the positive comments and valuable suggestions. The significant number of typos and formatting errors in the figures have been corrected in the Supplemental materials, please refer to the Supplemental Figure 1-Supplemental Figure 4.